# Digital Repeat Photography Application for Flowering Stage Classification of Selected Woody Plants

**DOI:** 10.3390/s25072106

**Published:** 2025-03-27

**Authors:** Monika A. Różańska, Kamila M. Harenda, Damian Józefczyk, Tomasz Wojciechowski, Bogdan H. Chojnicki

**Affiliations:** 1Laboratory of Bioclimatology, Department of Ecology and Environmental Protection, Faculty of Environmental and Mechanical Engineering, Poznań University of Life Sciences, ul. Piątkowska 94, 60-649 Poznań, Poland; kamila.harenda@puls.edu.pl (K.M.H.); damian.jozefczyk@up.poznan.pl (D.J.); bogdan.chojnicki@puls.edu.pl (B.H.C.); 2Department of Biosystems Engineering, Faculty of Environmental and Mechanical Engineering, Poznań University of Life Sciences, ul. Piątkowska 94, 60-649 Poznań, Poland; tomasz.wojciechowski@up.poznan.pl

**Keywords:** digital repeat photography, flowering, *k*-nearest neighbors, phenology

## Abstract

**Highlights:**

**What are the main findings?**
The *kk*nn method empowers the phenocamera-based flowering time determination.The effectiveness of the developed methodology depends on the plant’s characteristics.
**What is the implication of the main finding?**
The presented method enables the definition of the start, end, and flowering duration.Phenocameras can enhance the efficiency of conventional phenological research.

**Abstract:**

Digital repeat photography is currently applied mainly in geophysical studies of ecosystems. However, its role as a tool that can be utilized in conventional phenology, tracking a plant’s seasonal developmental cycle, is growing. This study’s main goal was to develop an easy-to-reproduce, single-camera-based novel approach to determine the flowering phases of 12 woody plants of various deciduous species. Field observations served as binary class calibration datasets (flowering and non-flowering stages). All the image RGB parameters, designated for each plant separately, were used as plant features for the models’ parametrization. The training data were subjected to various transformations to achieve the best classifications using the weighted *k*-nearest neighbors algorithm. The developed models enabled the flowering classifications at the 0, 1, 2, 3, and 5 onset day shift (absolute values) for 2, 3, 3, 2, and 2 plants, respectively. For 9 plants, the presented method enabled the flowering duration estimation, which is a valuable yet rarely used parameter in conventional phenological studies. We found the presented method suitable for various plants, despite their petal color and flower size, until there is a considerable change in the crown color during the flowering stage.

## 1. Introduction

Plant phenology is the study of the timing of plant seasonal development and its relationship with environmental factors, both biotic and abiotic [1]. There are various scientific disciplines where plant growth stage studies are applied, e.g., bioclimatology, agronomy, and ecology [2]. The interest in phenological observations is growing [3], which is linked to the continuous progress of climate change leading to possible shifts in plants’ growth stages [4]. Flowering is the simplest phase to spot for both visual distinctiveness and the rapid character of changes in plant shoots, facilitating the monitoring of climate change’s impact on plants. Anthesis is a key period in the reproduction cycle, ensuring fruit and seed development, which are essential for wildlife and humans. Pollinators can benefit from nectar and pollen in exchange for pollination, enabling many plants to exchange genes and reproduce. While various species show different levels of climate sensitivity, changes in the flowering time can lead to mismatches between pollinators and plants [5]. Moreover, varying responses to climate change can increase the risks of alien species invasions as flowering determines the timing of the following growth stages, enabling alien species to close their reproduction cycle earlier and change their distribution.

At the beginning of modern phenology study, plants were observed by trained researchers, who recorded the dates of phases such as leaf development, inflorescence emergence, flowering, or senescence [2]. They used standardized guidelines and descriptions of phases of interest within a network, but these usually vary between countries. Despite attempts to create universal coding for phenophases (like the BBCH scale [6]), even recent studies use various descriptors for analyzing the growing stages (compare: [7,8]).

In recent decades, with the development of monitoring methods such as satellites [9] and phenocameras [10], plants can be observed remotely without or with limited laborious field studies. The satellite approach is focused mainly on a broader spatial scale than conventional field observations performed by individuals in a limited area, thus requiring large human resources to obtain observations on a regional scale. Nevertheless, satellite observations are usually restricted to distinguishing phases as the start, peak, and end of the season at the plant community level [11]. This arises from low temporal resolution, data contamination (caused by, for example, clouds or aerosols), or spatial resolution. Researchers sometimes use the abovementioned method to investigate more precise growing stages, such as flowering [7,12], but this is currently only for relatively homogenous land cover areas.

Near-surface remote sensing (digital repeat photography) was developed to address the spatial and temporal resolution constraints in remote plant monitoring [10]. While used mainly in sites measuring matter and energy fluxes [3], where phenological characteristics such as the start or length of the season are required, phenocameras enable more precise observations depending on the distance of the camera from the observed vegetation, taking into consideration the individual plant [10] and/or individual growing stage. While an increasing amount of research is focused on phenocamera usage, both the typical ‘landscape level’ frame and specific growing stages are rarely considered, leading to more general than precise phenological observations [13,14,15]. When studies are conducted to observe the precise growing stages, as in traditional phenological studies, they are constructed specifically for this case with a camera near the observed plant [8,16]. This is causing the need to utilize more cameras for multiple plant observations while the phenological information for the plant canopy is missing. In an era of rapid machine learning (ML) expansion, numerous studies apply ML to facilitate image classification or recognition [17].

This study aimed to develop an easy and efficient way to indicate the flowering of 12 deciduous woody plants in a typical phenocamera image time series (landscape level). A basic machine learning algorithm was trained on one year of images to create separate models for each plant to achieve this. Ground truth data obtained within periodic field observations of each woody plant were used in the training instances. The models were then tested on another set of whole-year image time series from the site for which the ground truth data were also available. The assessment of the model’s performance was intended to answer several questions:(a)Can flowering be distinguished from other vegetation statuses based on changes in the RGB indices in a plant canopy-level (landscape) resolution monitoring camera?(b)How do the models’ classification performances differ between plants with various flowering patterns?(c)What are the constraints of the developed method?


## 2. Materials and Methods

### 2.1. Study Site

This study was conducted in the Dendrological Garden of Poznań University of Life Sciences (PULS, located in Greater Poland voivodeship, Poland). Arboretum, founded in 1922, is located in a highly urbanized area of Poznań city. Still, as a part of the green wedges designed in the 1930s, the garden is partly surrounded by other green spaces. It is situated on post-glacial sandy deposits and river accumulation sands (fulfilling anthropogenic soil criteria at present), with an elevation of 70–80 m above sea level [18].

The average annual air temperature (Ta) for the Poznań area is 9.4 °C, and the average annual sum of precipitation (P) equals 539 mm, for the reference period 1991–2020. Considering the reference period, the two years analyzed in this study were warm (Ta = 10.9 °C in both 2022 and 2023) but they varied in precipitation greatly. The year 2022 can be assessed as dry, with P = 418.8 mm, and 2023 as humid with P = 710.5 mm (Institute of Meteorology and Water Management (IMGW) data [19] for precipitation, air temperature values derived from the meteorological station at the site).

#### Analyzed Plants

The camera frame covers one of the oldest parts of the garden, with a collection of woody plants, both locally occurring and of foreign origin (Figure 1). In the foreground, approximately 20 m from the camera, there is a lawn area with one of the observed plants: *Magnolia salicifolia* (Msa). In the distance 20–50 m from camera, most of the observed plants are present: three Rosa genus shrubs (*Rosa glauca*—Rgl, *Rosa spinossisima*—Rsp and *Rosa* × *pteragonis*—Rpt), lilac *Syringa vulgaris* (Svu), small-leaved linden *Tilia cordata* (Tco), single-seeded hawthorn *Crataegus monogyna* (Cmo), cherry plum *Prunus cerasifera* (Pce), common persimmon *Diospyros virginiana* (Dvi), and the tree of heaven *Ailanthus altissima* (Aal). The other two plants observed in this study were Norway maple *Acer platanoides* (Apl) and sycamore *Acer pseudoplatanus* (Aps), the distance of which from the camera is approximately 100 m. All these plants differ in the color and size of their flowers, the flower location on the shoots, and the order of seasonal flowering and leaf development (Table 1).

### 2.2. Digital Repeat Photography

#### 2.2.1. Camera

The phenocamera was installed on 4 December 2020, on the first floor of one of the PULS buildings (N 52.426749°, E 16.895685°). The garden’s location in relation to the building forced the camera lens to face south, which is not a preferable direction for digital repeat photography due to the influence of direct radiation on the colors recorded [8,20].

The 4Mpx IP camera’s (model Dahua Technology Co., Ltd., Hangzhou, China) focal length was set at 2.7 mm with a field of view of 104º. Images are continuously taken at 10-min intervals for 24 h a day, leading to numerous monochromatic night images and daytime images in color. Every image has a size of 2560 × 1440 pixels with a resolution of 300 dpi, and its time is set to Coordinated Universal Time (UTC). The camera frame is fixed, meaning that every image presents the same fragment of the Dendrological Garden.

#### 2.2.2. Images

The camera collected images throughout 2022 and 2023, allowing complete information on the vegetation’s state during the two analyzed years.

For each of the 12 analyzed plants, the regions of interest (ROIs) were defined (Figure 1). The ROIs were determined on the largest area of the image within the fragment of the analyzed plant, with attention paid to the possible seasonal partial plant overlay by neighboring species. Moreover, the definition of the ROIs considered possible shifts of the plant branches and shoots due to weather conditions (wind, rainfall) and increased biomass during the season.

For the ROI areas within every image, the following color indices (features) were calculated:digital numbers (dn) for red (R), green (G), and blue (B)—with the values scale from 0 to 255,brightness (BRI)—calculated as the sum of the RGB digital numbers (1),BRI = R_dn_ + G_dn_ + B_dn_(1)

relative values of RGB (RI, GI, BI)—with a scale between 0 and 1 (2),


(2)
$${\mathrm{Col}}_{i}=\frac{{\mathrm{Col}}_{\mathrm{dn}}}{R_{\mathrm{dn}}+G_{\mathrm{dn}}+B_{\mathrm{dn}}}$$


green excess index (GEI) [21] (3),

GEI = 2 G_dn_ − (R_dn_ + B_dn_)(3)

standard deviations for all of the above.

As a result 16 features describing every plant ROI throughout the time series were created.

The ROI definition and RGB indices were calculated using the ROI-averaged approach described in Filippa et al. [22] and the *phenopix* (version 2.4.4) package [23] in the R programming language (version 4.4.0) [24].

### 2.3. Ground-Based Observations

Parallel to the phenocamera recording, the conventional phenological observations were carried out in the following periods: 15 March–7 October 2022 (52 times) and 22 March–28 September 2023 (47 times). The frequency of the observations was related to the occurrence of the flowering phase of the observed plants. Each field visit consisted of visual observations supplemented with the handheld photography of studied plants. A photography comparison allowed for a more precise plant phase assessment [25].

The plant flowering dates were determined based on field observations using the BBCH scale [26,27]. The BBCH scale was designed to be universal for various species. For many crops and some genera, there are specific versions of the scale [26,28]. The two-digit code created from digits 0–9 informs about the principal growth stage (first digit) and its progress (secondary growth stage, last digit). Flowering is indicated by code 6. The beginning of the flowering phase is usually marked as 60 [2] or 61 [4,7], where code 1 stands for 10% of open flowers. A greater number in the secondary code refers to the subsequent stage of development within the principal growth stage. Thus, 65 indicates full flowering (>50% open flowers), 67 means that most of the flowers have dry or fallen petals, while 69 describes the end of flowering with visible fruit structures [27].

To prepare the phenological data for the ML analysis, the flowering stages of the observed plants were marked between BBCH 61 and BBCH 67. Phase BBCH 61 is often used as a blooming start instead of BBCH 60 [27] due to the more robust certainty of the observations, as spotting the first open flowers can be extremely difficult in the case of large woody plants with no access to the whole crown. Phase BBCH 67 was selected as the end of the flowering stage due to the growth stages overlapping with the leaf (Apl, Msa, Pce) or fruit development (Aps, Cmo, Rgl, Rpt, Rsp, Svu, Tco). Aal and Dvi are dioecious plants, and both that are present in the camera frame are male; thus, there is no overlap of the growth stages (plants have fully developed leaves before flowering). However, these plants lose their flowers immediately after pollination, which leads to bare panicles/stems resembling non-flowering stages in the late flowering stage. The application of the BBCH 61–67 flowering period also prevents uncertainties related to the frequency of field observations. Since they were carried out at a non-daily frequency, the early signs of the start and end of flowering could be overlooked.

### 2.4. Machine Learning

#### 2.4.1. Preprocessing

Due to the presence of night images and the garden’s commercial activities, including evening light shows in the autumn-winter periods of the analyzed years, the color indices data were preprocessed according to the local sunrise and sunset schedules, attraction opening hours, and lastly, the values of the standard deviation of the color indices equaled to zero (which was the sign of a too dark ROI area). As the images from these periods had an unpreferable exposition (too dark at night periods and too bright, with unnatural colors in periods of commercial activity caused by artificial lights), they were removed from the datasets.

#### 2.4.2. Algorithm Selection

Preprocessed data containing 16 numeric features (indices based on averaged RGB channels in the ROI, see Section 2.2.2) were the base for selecting the most adequate ML algorithm for the flowering classification. Data were labeled with binary classes: positive (flowering period) and negative (non-flowering period). The positive class (flowering) was marked for every image RGB index within phases BBCH 61–67, according to the observed stages at the research site. Then, the baseline model and five simple and computationally efficient classification algorithms were chosen for benchmarking: baseline algorithm that assigns a higher frequency class to all the images (featureless), weighted *k*-nearest neighbors (*kk*nn), support vector machine (svm), random forest (ranger), naive bayes and classification tree (rpart) using the dedicated to ML purposes R package, *mlr3verse* (version 0.2.8) [29]. A subsampling cross-validation method was applied to evaluate the algorithms’ performance, with 75% of the randomly selected image indices from the one-year time series used as training sets and the remaining 25% as test datasets. This procedure was repeated 10 times to assess the variability of the evaluation results. The subsampling method was a basis for learner selection and the further validation processes described below (*k* value, kernel, and feature selection).

Each algorithm’s performance was evaluated using the recall (REC), precision (PREC), and Fβ score (Fβ) measures calculated from the confusion matrix. Recall is a score for true positives TP in all the positive observations, including false negatives FN (4).(4)$$\mathrm{REC}=\frac{\mathrm{TP}}{\mathrm{TP}+\mathrm{FN}}$$

It is an important measure in the case of unbalanced dataset classes [30], where the main focus of the model is to predict as much as possible of the target class, and the cost of having false positives is low.

On the other hand, precision is a score of true positives in a set of all positive responses, including false positives FP (5),(5)$$\mathrm{PREC}=\frac{\mathrm{TP}}{\mathrm{TP}+\mathrm{FP}}$$
meaning it is not focused on the problem of missing out on the target class but rather on the current identification of this class, with the high cost of false positives. In the case of the image time series containing stages from the whole growing season, where flowering is a relatively short phase with a low representation within the dataset, excluding models with tendencies to omit the target class predictions was more important than allowing for false positives. Thus, REC was the main measure to evaluate the models’ performance in the study. PREC served as an additional measure.

However, because PREC tends to decrease with increasing REC and vice versa, the trade-off for these measures was necessary for the algorithm selection process, especially since some of the algorithms reached high REC values with a very poor performance in the PREC measure. As low PREC signaled that the models could not distinguish flowering from non-flowering phases, the Fβ score was calculated for benchmarking, allowing the assessment of both measures with one value (6).
(6)$$F\beta =2\times \frac{\mathrm{PREC}\times \mathrm{REC}}{\mathrm{PREC}+\mathrm{REC}}$$

In all 12 datasets selected for training, the best performance reached the weighted *k*-nearest neighbors algorithm (Table 2); thus, further analysis applies to this classification method.

#### 2.4.3. Flowering Stage (Target) Designation for Training Purposes

Phenological field observations are prone to error due to the observer’s subjective decisions and experience [2]. Even relatively precise BBCH scale observations are based on the visual assessment of the percentage of stage development that can be difficult to achieve accurately with large perennials, like trees, whose crowns are far from the observer. Taking those limitations into account, there were three following data treatments applied to assign the binary class labels (Figure 2, Table 3):

A. Observer’s perspective—based on the decision to distinguish the flowering phase for stages between BBCH 61 and BBCH 67, while transitional stages, BBCH 60 and BBCH 68 to BBCH 69, are marked as non-flowering stages. In case of the absence of observations from stages BBCH 61 and BBCH 67, the flowering classes were extrapolated due to the number of days with missing observations within two known stages and the corresponding observations in other years with field observations to complete the BBCH 61–67 flowering period.

B. Uncertain data removal—this approach was an attempt to cope with the possible misinterpretation of observed stages, leading to the deterioration of the classification results. Removing data from days of year (DOYs) between the last non-flowering observation and the first observation in the range of the BBCH 61–67 flowering stage (thus some BBCH 60 observations were removed as well) and between the last observations within this range and the first non-flowering observation after blooming (which exclude observations in the range of BBCH 68–69) produces data with only the most certain observations remaining. Thus, selected training datasets contained a reduced amount of the ROI’s image color indices data.

C. Semi-supervised learning for label determination—since treatment B does not solve the problem of extremes in flowering stages because it cancels the ability of the algorithm to ‘learn’ from the transitional stages, a third approach to data treatment was applied. It used the *kk*nn algorithm and small training datasets for each plant in semi-supervised learning. The dataset contained only the days of the flowering period and a range of 10 days before and after the first and last non-flowering stage observations, respectively. Those preceding and following flowering 10 days were used as training sets for the non-flowering classes since the color indices during this period were relatively similar to extremes of the phase. Training datasets for flowering classes were formed from days including phases 62–67 (when the specific plant’s observations did not consist of some of the phases, the training data included the period from the former to latter observations from this range, for example, 63–66). Stage 62 was selected to confirm if the 61 stage is similar enough to the rest of the stages in the training sets. The test sets included two subsets of data from before and after the flowering training sets. Learner hyperparameter selection, training, and prediction methods are described in the next section of this paper.

A comparison of the flowering classes to DOYs obtained with A, B, and C treatments is presented in Figure 3.

#### 2.4.4. Kernel (Weighted) *k*-Nearest Neighbors Method (*kk*nn)

The kernel *k*-nearest neighbor algorithm (weighted *k*-nearest neighbors) is one of the simplest and most well-known classification techniques in ML, based on identifying nearest neighbors, that is, the closest points in the *n*-dimensional space according to some distance measures (kernel functions), and assigning to a given point the class of the majority of those neighbors [31,32]. It is a memory-based classification often named the ‘lazy learning’ technique since the algorithm just memorizes the training dataset and compares the new data with it.

#### 2.4.5. Training Datasets

The general strategy was to train the *kk*nn algorithm on the data from a one-year time series and then test the obtained model on another whole year of data unknown to the algorithm. However, the frequency of field observations in 2022 was more effective in terms of precisely determining the transitional stages (especially flowering onset) for eight plants: Apl, Cmo, Msa, Pce, Rgl, Rpt, Rsp, and Svu, while those stages for four plants (Aal, Aps, Dvi, Tco) were more accurately identified in 2023. Thus, there were eight training sets from 2022 and four from 2023. Each plant training set was labeled using the three data treatments described above: A, B, and C. The date and DOY information were not used for the algorithm training in any data treatment.

To tune *kk*nn models, each of the previously prepared datasets was transformed using the following transformation categories:no transformations (nt)—none of the data were transformed, which means that all the data and all 16 features were used in the algorithm training,feature selection (fs)—individually calibrated modification in training sets to achieve the best set of features (color indices) for the flowering classification (Table 4),data filtering (df)—the training and test datasets were filtered to remove 50% of the diel image RGB parameters with a green excess index standard deviation (gei.sd) value lower than its diel median. The gei.sd was chosen since this feature combines information from all three color (RGB) channels, and when the image is biased due to unfavorable camera exposure conditions, e.g., shadowing, filtering out lower diel gei.sd values usually enhances RGB dataset quality,both feature selection and data filtering (fs.df)—each plant dataset was transformed using both feature selection (fs) and data filtering (df).

As a result, 3 × 4 (data treatments A, B, C, and transformations nt, fs, df, fs.df) datasets were prepared for each of the 12 woody plants analyzed in this study, creating 144 training datasets for future models.

#### 2.4.6. *k*-Value and Kernel Determination

For datasets A.nt, B.nt, and C.nt, the values of *k* (that is, the number of ‘neighbors’, closest points in the *n*-dimensional space, where *n* stands for a number of features, which equals 16 in this case) and kernel (weighted method to determine the ‘closest’ neighbors, 10 various to choose, see: [31]) were established using the subsampling method mentioned earlier (Section 2.4.2). The *k*-value tuning range was set to 3–15, where only odd values were applied to eliminate the tie in the classification results. The range limits were set due to the standard procedure, where *k* = 1 is eliminated from model tuning to prevent the singular closest neighbor from deciding about the predicted class, and *k* = 15 is a slightly higher value than the square root of the total number of diel images. The kernel and *k*-value were determined pairwise (7 *k*-values and 10 kernel methods) for each plant dataset using the *mlr3verse* package [33]. The tuning terminator was set to stop after 30 iterations without progress in the recall values (threshold progress = 0). The best kernel and *k*-values regarding the recall measure were determined since, with numerous data, the need to minimize false positives was lower than the ability to predict the flowering class. The kernel and *k*-value determined for A.nt, B.nt, and C.nt were then used in models of every type of transformation in the given approach (Figure 4).

#### 2.4.7. Test Datasets

After fitting the models to the training data by determining the best *k*-value and kernel, the models were tested using test datasets. These data were neither used in the validation process nor in model fitting. For Apl, Cmo, Msa, Pce, Rgl, Rpt, Rsp, and Svu, the plants camera observations were from 2023, and for plants Aal, Aps, Dvi, and Tco, they were from 2022, enabling the assessment of the efficiency of flowering stage identification using the developed models.

The models’ predictions for every image RGB indices were then converted into diel classifications, since each plant flowering stage record is usually carried out on diel resolution [2]. The following equation was used to designate each DOY of the flowering stage where a 50% threshold of positively labeled classes was applied (7):(7)$${\mathrm{CL}}_{\mathrm{DOY}}=\left\{\begin{array}{c} 1\;\mathrm{if}\;\frac{P\times 100}{P+N}\geq 50\%\\ 0\;\mathrm{if}\;\frac{P\times 100}{P+N}<50\% \end{array}\right.\;$$
where:

CL_DOY_—flowering stage class (1—positive, flowering or 0—negative, non-flowering),

P—number of diel positive predictions,

N—number of diel negative predictions.

The following criteria were used to determine each plant’s best model performance:onset day shift (ODS);ODS = OD_pred_ − OD_obs_(8)where:

OD_pred_—predicted onset day [DOY],

OD_obs_—observed onset day [DOY];

the flowering days share (FDS),
(9)$$\mathrm{FDS}=\frac{{\mathrm{FD}}_{\mathrm{pred}}\times 100}{{\mathrm{FD}}_{\mathrm{obs}}}$$where:

FD_pred_—predicted flowering days

FD_obs_—total number of days in one of three observed periods: BBCH 61–67, extremes of the flowering phase, and non-flowering stages (Figure 5).

## 3. Results

### 3.1. Field Observations

Flowering stage observations for the analyzed plants are presented in Appendix B (short version in Table 5). The mean time break between the observations performed in the field was 3 days in 2022 and 3.1 days in 2023; however, the increased observation frequency concerned various months in both years.

Comparable thermal conditions in both years of field observations resulted in a very homogeneous flowering start schedule of the observed plants. The earliest (April) were species that bloom before left unfolding: Pce, Msa, and Apl. In May, the flowering of Rpt, Svu, Rsp, Cmo Aps, and Rgl started. Late flowering plants in the camera frame were Dvi, Aal, and Tco.

Generally speaking, the blooming period of the ornamental plants (Msa, Svu, Rpt, Rsp, Rgl) was longer than that of other plants, except for Apl, whose flowering was prolonged likewise. The most significant difference between the blooming duration between 2 years was observed in Msa. It had a noticeably more extended flowering period in 2022 than in 2023 due to frost damage in the latter year. Additionally, the secondary blooming of roses (Rgl, Rpt, and Rsp) was observed during autumn 2023. However, their blooming events visually resembled stages not higher than flowering extremes (BBCH 61 or BBCH 68–69) of the primary flowering in the earlier part of the season.

Plants also differed in anthesis synchrony. Numerous Cmo flowers started to open at a similar time on the whole crown; thus, distinguishing phase BBCH 61 was hindered even with frequent observations. In contrast, Rgl’s flowering pattern includes a slow transition from the start to the peak of flowering (which appeared 2 weeks after blooming started) and a more rapid change from the peak to the end of flowering.

### 3.2. Classification Efficiency

Target designation and data transformations led to the 144 versions of training datasets that were the inputs for the *kk*nn models. As described above, three measures were applied to assess each developed model performance: REC, PREC, and Fβ. The REC was assumed to be the most adequate measure of the model’s performance when the target positive class is significantly less numerous than the non-flowering class. Thus, the model-fitting process was based on achieving the highest REC value. However, Figure 6 also contains precision and Fβ score values, as they are relevant in the model performance assessment.

There are significant differences between the REC values of each plant model. The results can be divided into three groups. The highest REC group (REC ≥ 70%) consists of the following plants: Svu (97.9–100%), Rpt (76.3–89.6%), Cmo (76.9–87.4%), Msa (81.3–86.8%), Pce (78.6–84.9%), Apl (70.8–84.1%), and Rsp (69.9–80.8%). In the second group, with medium recall values (REC 45–70%), the following plant models were ranked: Rgl (47.6–69.7%) and Aps (46.1–60.8%). The third group, where none of the models exceed REC = 50%, contains models for plants: Aal (23.7–48.3%), Dvi (27.1–41.1%), and Tco (18.2–25.6%).

The PREC values were significantly lower than the REC values in the majority of models. While PREC was not the main focus in fitting the hyperparameters, values over 50% for this measure were reached by all models of Pce (68.5–80.2%) and some models of Msa, Rpt, Svu, and Cmo. The lowest precision values characterize models for Aal, Apl, Tco, and Dvi, with the precision value for the latter not reaching over 14% in any model.

The Fβ score (joining results for the measures mentioned above) for models achieved a wide range of values, with highest scores for Pce, Msa, Svu, Rpt, and Cmo (45.4–82.4%), medium for Rgl, Rsp, Apl, Aps, and Aal (25.0–54.6%), and lowest again for Dvi and Tco (below 25%).

### 3.3. Diel Classifications

While the REC and PREC values give a general idea of the model’s performance, only classifications summarized by days show clearly how accurate the models were in predicting the exact days of flowering stages for observed plants. The best-fitted models were chosen due to the diel classification results (Appendix A), and the further description is related only to them (Figure 7 and Table 6).

The most common phenological observations include the onset of the phase [34]. Figure 7 presents the predicted beginnings of flowering in relation to the observed ones. Models of Cmo and Pce had no onset shift, while Aps and Tco had the most notable shift (5 days in the absolute value).

Unlike field observations, models enable the depiction of the duration of the flowering phase. Daily flowering predictions for Svu were the most accurate, with all (FDS = 100%) the observed days of flowering assigned to the positive class (Table 6). The Apl, Aps, Cmo, Msa, Pce, Rgl, and Rsp models reached an FDS between 86.7% and 100% (1–3 days misclassified). The plants with the poorest diel performance were Aal, Dvi, and Tco, with less than 50% of flowering days predicted by the models.

Additionally, models for Aal, Cmo, Pce, Rpt, Rsp, Svu, and Tco did not confuse any non-flowering days with flowering predictions (FDS = 0%). The Apl, Aps, Msa, and Rgl models assigned 2 and 3 days from this period as a flowering stage (FDS 0.7–1%), but the exact DOYs differed between plants. Msa and Rgl incorrectly predicted images with snow cover as flowering (DOY 331, 336–337, and 330–332 in Msa and Rgl, respectively), while the Apl model assigned days from early leaf discoloration (DOY 258 and 298). The Aps model misclassification (DOY 131–133) was related to stage BBCH 5 (inflorescence emergence). The Dvi model misclassifications during the non-flowering period reached 18 days (FDS 5.4%), even 2 months after the designated flowering stage.

The periods between the last observation before flowering (including BBCH 60) and BBCH 61, and between BBCH 67 and the first observation after any signs of the flowering phase (including these from phase BBCH 68 or 69, Figure 5) served as additional parameters in the models’ performance assessment (Table 6, flowering extremes). The Cmo, Msa, Rsp, and Tco models made no mistakes in classifying DOYs in these periods (FDS = 0%), and the Aal, Dvi, and Rpt models assigned one day from the extremes to the flowering class (FDS 5–11.1% depending on the period length). In other models, the flowering class was assigned 3 (Pce, Rgl, FDS = 10.7–25%), 5 (Aps, Svu, FDS = 35.7%), or 7 times (Apl, FDS = 58.3%). The FDS of the flowering extremes shows that more detailed models’ performance as misclassifications of the phase extremes can be treated as correct in contrast with, e.g., DOYs indicating fruit ripening or dormancy. For Svu and Apl, those positive classifications show extremes of BBCH 6 but in the Aps model, it is the continuation of the classifying stage BBCH 5 as flowering.

## 4. Discussion

### 4.1. Models’ Performance

In the presented study, we used various data and model-tuning approaches to obtain predictions of flowering in 12 deciduous woody plants that most closely reflected conventional field observations.

#### 4.1.1. Data Treatment

When no additional tuning was applied, various target designation methods (A, B, or C) did not give an overall better performance in the REC values. According to the REC, data treatment A was the most suitable for Aal and Tco, while the least was for Apl, Aps, Dvi, Rgl, and Rpt (Table A3). Removing data from the start and end of flowering (data treatment B) was helpful in the cases of Cmo, Msa, Rgl, Rpt, Rsp, and Svu plants, but resulted in the lowest REC of Pce and Tco. The semi-supervised (C) data treatment resulted in the highest REC for Apl, Aps, Dvi, and Pce (for Aps, Dvi and Pce C.nt gave the best performance of all), while the lowest was for Aal, Cmo, Msa, Rsp, and Svu.

#### 4.1.2. Model Tuning

Feature selection (fs) applied to each data treatment did not robustly influence most models’ performances. However, Apl and Cmo noticed a slight increase in the REC values, while for the Aal, Dvi, Msa, and Svu models, feature selection brought a deterioration in the REC values for all three data treatments. Data filtering (df) had a generally stronger impact on the model’s performance than feature selection (fs). Filtering positively affected all the treatments for plants such as Aal, Apl, Msa, Svu, Tco, and Rgl. In the case of Rgl, the REC has been raised by about 10% due to df. For the Cmo and Rsp models, REC values drop every time after gei.sd filtering.

The simultaneous application of feature selection and data filtering (fs.df) either enhanced or reduced the positive impact of a single (df or fs) model-tuning process in most cases. This transformation increased all the REC values for three data treatments of Apl and Rgl and caused REC decreases in Rsp models.

#### 4.1.3. Joint Effect of Data Treatment and Model Tuning

The joint effect of data treatment and model-tuning analysis indicates that treatment A was sufficient enough to reach the highest REC of all the treatments in only two cases: Aal and Tco. However, both are characterized by low REC values in any model applied. Treatment B, jointly with model tuning, improved the models’ performance (by reaching the highest REC values) for five plants: Cmo, Msa, Rgl, Rsp, and Svu. Semi-supervised treatment (C) and tuning improved positive class identification in six plant models: Apl, Aps, Dvi, Pce, Rpt, and Svu. The Svu models B.df, B.fs.df, and C.fs.df were characterized by REC = 100%.

Considering various models’ performance with respect to the morphological characteristics of the analyzed plants, e.g., flower abundance, petal colors, and inflorescence placement on plant stems (Table 1), a separate model adaptation to each analyzed plant may be required in future method application.

#### 4.1.4. Diel Classification Results

The effectiveness of the diel classifications, which can be compared with conventional observation, varies among the models. Taking into account the frequency of field observations (usually once or twice a week), most models (Aal, Apl, Cmo, Dvi, Msa, Pce, Rgl, Rpt, Rsp, and Svu) showed shifts from the onset date that were not higher or equal to the possible error resulting from the frequency of the observations.

The described methodology utilized two classes and was not designed to designate the peak of flowering, another commonly recorded phase in field phenological observations. However, the flowering duration is even more informative than another single point in time, as it can serve as data that are useful in a study of available food resources for pollinators, a plant’s condition, or its plasticity to environmental factors [5,34]. In this study, models for nine plants—Apl, Aps, Cmo, Msa, Pce, Rgl, Rpt, Rsp, and Svu—can predict the flowering duration from the date of the actual beginning (0–3 day shift) to the end of flowering, in some cases equal to the observed BBCH 67 (Cmo, Msa, Rgl, Rpt, and Rsp models), in others extended to late signs of flowering, as phases BBCH 68–69 (Apl, Aps, Pce, and Svu models).

As climatic factors have changed rapidly in recent years, with a significant increase in air temperatures, which is the main factor in advancing plant development timing [5,35], a deeper understanding of plant phenology at both the individual and population levels is suggested by multiple authors [5,34,36]. Through the presented method, flowering period designation (with its onset date, flowering termination, and duration) is more reachable than with time-consuming field observations. In addition, these parameters are far more accurate regarding the plant response to climatic factors than one point in time indicating when the flowering begins, to which conventional phenology is often limited [34].

The Aal, Dvi, and Tco models have the poorest performance in predicting flowering duration, with less than half of DOYs (or even none). However, CL_DOY_ threshold value manipulation (lowering to 40%) enabled classification improvement for the Aal model, which resulted in a significant increase in predictions: from FDS 42% (5/12 days) to 75% (9/12 days). At the same time, the number of misclassified days rose by only one, but in a period of flowering extremes. Such threshold manipulation was not effective in the Dvi and Tco models due to inconsistencies in the percentage of correctly classified images even at the peak of flowering and a high rate of incorrect positive class predictions long after the end of the flowering stage. Nevertheless, even with a few days offset or missing from predictions, consistent phenological information obtained automatically can still provide valuable data about long-term trends in flowering time, since the first increase in the number of images classified as flowering occurs during the observed flowering.

### 4.2. Limitations

#### 4.2.1. Plant Characteristics and Location

The models’ performance seems to be dependent on plant visual characteristics. The flowers or inflorescence of ornamental plants (Cmo, Msa, Pce, *Rosa* sp., or Svu) are clearly visible due to the distinguished petal color, size, and flower abundance, which caused the contrast to their overall crown color in comparison with leaf verdancy and resulted in a more robust change in the RGB parameters. Plants with flowering emergence before leaf unfolding (Apl in addition to Msa and Pce) have flowering that is contrasting with the crown’s bare branches. For the above-mentioned plants, the models’ application enables effective flowering predictions and allows for the future reduction or elimination of ground observations since the accuracy of automatic observations is comparable with the observer’s reports. The Aal, Dvi, and Tco models proved much less sufficient in flowering predictions. All three plants have flowers hidden between the leaves (Dvi) or small, greenish flowers (Aal, Tco), making no contrast with the crown color from the phases before and after flowering. Hence, plants with more subtle flowering patterns will limit the effectiveness of this method.

Plant distance from the camera may affect the predictions in a limited way. The Cmo, Msa, Pce, Rpt, Rsp, and Svu models, with good prediction results, are located relatively close to the phenocamera (20–50 m). Yet, Apl models resulted in good predictions even though the plant grows over 100 m from the camera and is partially covered by vegetation located closer to the camera. Further-located plants are also usually partially covered by neighboring plants. Nevertheless, the plant fragment depicted on the images was enough to predict the flowering of Apl, Cmo, and Pce.

#### 4.2.2. Observer’s Bias

Supervised machine learning involves observer-labeled data, on which algorithms then “learn” to identify patterns that are difficult to distinguish using simple statistical analyses. Therefore, successful flowering classification in this study depended on field observations and the observer’s correct plant phenophase identification. However, field observations are known for being prone to bias according to the observer’s experience and/or even preferences. An assessment of the growth stage is subjective and may differ between observers, and it occurs most often in the case of extreme stages [25]. Some studies deal with this issue by sending several phenological observers to assess the plant growth stages [37].

Field observation frequency is not strictly specified [2,25], and it varies between research (e.g., compare [38] and [8]). In this study, observations carried out at 2–4-day intervals were sufficient to determine the dates of the beginning and end stages of the flowering of individual plants. Reducing the number of observations could decrease the quality of the field phenological data, especially if the observation gap occurred when the first flowers opened (phases BBCH 60 and BBCH 61). These situations happened in the first year (2022) of observations for Aps, Dvi, Tco, and Aal. Thus, the ground-based data collected in the next year (2023) were applied to determine more precise labels in the training sets for these four plants.

Field observations are also limited due to the observer’s position relative to the plant. In tall tree studies, when both distance and the flowers’ small size can bias the observations (Apl, Aps, and Tco in this paper), even the use of binoculars can be insufficient for observing the first signs of flowering (BBCH 60 and BBCH 61).

The traditional phenological observations are separated in time. Plant phenological events are continuous processes, and the transition from one phase to another can occur within hours rather than days. Then, reporting ground observations, even daily, may not reflect the actual date of the phase. That considers mainly the phase onset and secondary growth stages (percentage of opened flowers). In this study, Cmo and Rsp were characterized by a dynamic transition of the subsequent flowering phases, in which the development of the flowering stage from 10% of open flowers (BBCH 61) to full flowering (BBCH 65) took place in less than 2 days. Determining the rate of the opened flowers itself is a difficult task for the observer. Therefore, field observations are often limited to the first signs of flowering [2,39]).

Despite the mentioned limitations, conventional field phenological observations are significant in phenology studies and provide ground truth data for any further remote sensing method. The observation’s robustness can be enhanced by handheld photography, allowing the comparisons between secondary phases, and by binoculars that help to spot flowers in hardly reachable parts of the plants. In the context of historical field observations, thanks to their continuity and long time series (reaching even to previous centuries and millennia), ground observations determine valuable information about changes in the plant response to abiotic factors in time [2,36] and at a level that, for new and more advanced methods, is still under development.

### 4.3. Phenological Methods Comparison

Digital repeat photography originated in modern color-based ecosystem phenology, where improvements in CO_2_ flux studies were needed. This application required phenological data about the vegetation season’s start, peak, and end. In a similar way, satellite data began to be used for this purpose in earlier years. Despite that goal, multiple attempts were made to go beyond vegetation indices in both satellites and RGB cameras. They concern the automation of phenological observations, including flowering.

Dixon et al. [12] detected the flowering of *Corymbia calophylla* trees on satellite data using a random forest regression algorithm and ground truth data derived from UAV images from an RGB camera. A satellite pixel resolution of 6 × 6 m allowed for flowering predictions for trees with clearly visible flowers that were abundant and contrasting with the crown greenness, with the purpose of predicting the spatial proportion of flowering.

The research by Dixon et al. is based on earlier attempts at flowering recognition in *Corymbia calophylla*. Campbell and Fearns [40] used ground images of the plant to detect white flowers using pixels with a parallelepiped algorithm. This approach was constrained by a minimal resolution of 10 pixels per flower to accurately distinguish flowering from the background (leaves, stems, branches).

In research by Nagai et al. [14], Mann et al. [16], Li et al. [41], and Taylor and Browning [15], digital repeat photography derived from phenocameras was used for the flowering analysis; however, the methodologies in those studies vary considerably. In the first one, where the tropical forest canopy was observed, and individual crowns were analyzed, no automation of phenology classification was applied.

In Mann et al. [16] and Li et al. [41], the deep learning method (CNN and YOLO, respectively) and phenocamera close-ups were used for the flower detection of small *Dryas* spp. subshrubs and *Camellia oleifera*, respectively. The first research allowed for the precise definition of the start, peak, and end of flowering; in second yield estimation was based on the detection of plant’s particular growth stages.

Taylor and Browning [15] presented the machine learning method for the broadest application with data from various agricultural PhenoCam network image time series [20]. VGG16, a deep learning model, was trained to classify dominant cover types, crop types, and phenological status, including flowering. The image data for training were labeled according to the visual inspection of the images themselves, not based on field observations. Thus, some phenological stages had to be converted into one, including tassels, flowers, and seeds development in the cereal crop type, which in the BBCH scale are stages BBCH5, BBCH6, and BBCH7, respectively.

A combination of ground truth data and digital imagery was the research subject of Nezval et al. [8] and Guo et al. [42]. In both studies, vegetation curves calculated from RGB channels for the close-ups of three tree species in floodplain forest [8] and summer maize field [42] were compared with the dates of the growth stages obtained by observers at the sites. The former research used the daily aggregated data of green chromatic coordinates for predicting phenology, and in the latter, all the color indices (also aggregated) and relations between pixels were considered.

The mentioned research depicts many possibilities to use and combine various phenological tools. With their resolution, satellites are more suitable for vegetation canopy research than individuals [12]. Satellite data can cover large areas but with a low resolution, and they suffer due to data noise caused by clouds and/or aerosol presence. RGB cameras, both fixed and movable, can obtain information about the canopy [14,15] or individual plants [8,14,16,39,40,41]. However, their cost, in terms of funding and labor, can increase robustly when the research requires multiple cameras [16] due to the zoomed view or drones needed to operate them [12]. The way of processing data is also an important issue, as the techniques using large computing power [12,16,41] can reduce the practical usefulness of the method.

Considering all the research above, the method described here presents a novel approach to digital repeat photography analysis. It allows for the utilization of inputs from every single image of the whole year time series, while extracting the ROIs’ averaged data on RGB channels. This is a characteristic simplification of the standard approach in digital repeat photography. While the need for standardization of the phenocamera technique is signalized [3], there are attempts to reduce the illumination conditions’ effect on the image time series [43] in the context of vegetation curves reflecting canopy greenness more accurately. However, these undesired types of interferences allow flowering detection, and smoothing noisy vegetation indices would blur the flowering information from the data. Applying a weighted *k*-nearest neighbors binary classification algorithm on simplified but not smoothed color indices is computationally efficient compared with deep learning and some other machine learning techniques, allowing the quick and easy performance of the phenological analysis.

### 4.4. Potential Applications

A presented combination of the methods can be easily applied at sites where conventional observations are carried out, e.g., botanical gardens and arboreta, where plenty of ornamental woody plants are usually present and in the vicinity of each other. Sites with a relatively high density of plants that the camera frame can cover should be suitable to enable a cost-effective comparison of plants from various habitats and with different sensitivity levels to changes in crucial climate factors in one location [4,38,44]. Orchards and nurseries are the other types of sites that can gain from utilizing the presented method. It can allow comparisons between varieties and can be further used to study varieties’ adaptation and resilience to climatic factors. This method uses RGB parameters and data representing the differences between image pixels during flowering caused by the contrast in flowers and the background colors (leaves, branches), expressed by the standard deviation of the RGB features. Hence, the method is worthy of testing in other ecosystems, such as crops, grasslands, and shrub communities, especially with a predominance of one species.

Digital repeat photography enables treating plant development in terms of continuous processes by recording changes in plant visual characteristics in a high frequency, which is particularly desirable in dynamically changing environmental conditions.

Applying the presented method to previously collected image time series, when calibrated using field-reported data, creates an opportunity to extend the long-term flowering study of the plants that are present within the camera frame. Moreover, digital repeat photography can provide a backup dataset for the eventuality of the observer’s importance in research sites beyond simple growth stage determination. Applying this method can potentially reduce the number of research visits, hence reduce human labor costs, and therefore increase the number of studied plant species.

## 5. Conclusions

This paper presents a novel approach to both species-specific phenological research and digital repeat photography. Weighted *k*-nearest neighbors algorithm application with proper target calibration and model tuning resulted in an innovative method for determining the flowering period without ground observations and excessive computing power. The demonstrated method is suitable mainly for ornamental plants and any others with flowers that are distinguishable from the background (leaves or branches) by color. It provides information about flowering onset and phase duration that is comparable with that obtained in field observations for those plants. These properties not only facilitate conventional ground-based phenological research but enrich it substantially. The presented methodology can be easily applied in institutions that are already conducting phenological observations, as they often report the dates of plants’ growth stages needed for target calibration.

## Figures and Tables

**Figure 1 sensors-25-02106-f001:**
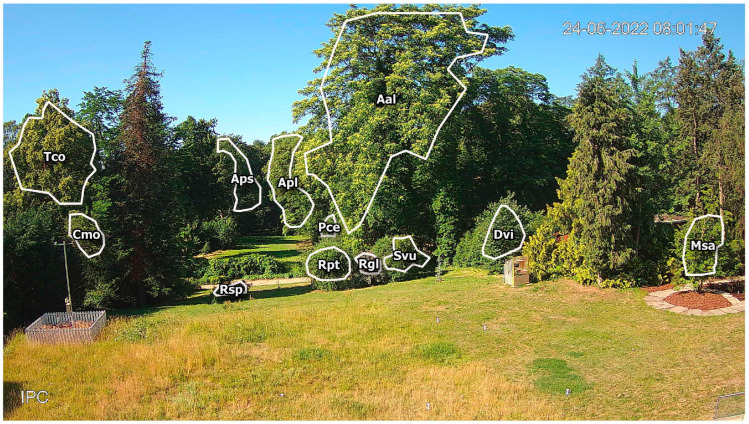
Sample camera frame image with outlined regions of interest (ROIs) with corresponding plant codes.

**Figure 2 sensors-25-02106-f002:**
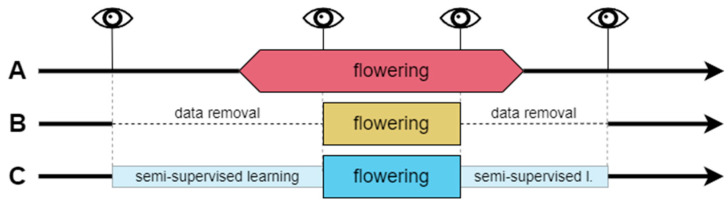
Data treatments (A, B, C) applied to determine the binary classes (flowering and non-flowering phases). Thick black lines represent the whole year’s timeline with RGB parameters of non-flowering stages. Wider boxes (pink, yellow, and blue) indicate the flowering stage class in every data treatment. Eye symbols and vertical lines indicate field observations. Horizontal dashed lines represent the absence of data. Narrow boxes (light blue) represent part of the data used in semi-supervised flowering classifications.

**Figure 3 sensors-25-02106-f003:**
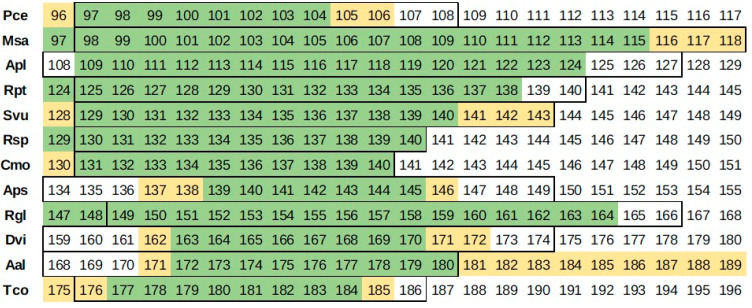
Comparison of the flowering day of year (DOY) determination according to approach A (yellow/light), B (green/dark), and C (outlined).

**Figure 4 sensors-25-02106-f004:**
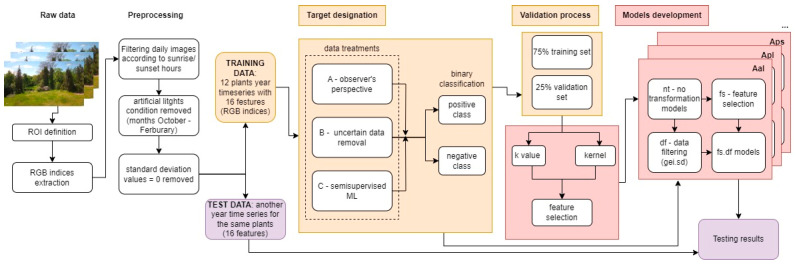
Machine learning process step by step. Arrows describe transitions to the next stage of data analysis, and colors indicate individual stages of the machine learning process (training—yellow, model development—red, testing—purple).

**Figure 5 sensors-25-02106-f005:**
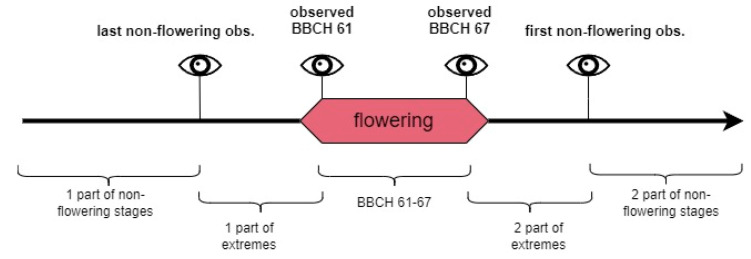
Three periods for flowering days share assessment.

**Figure 6 sensors-25-02106-f006:**
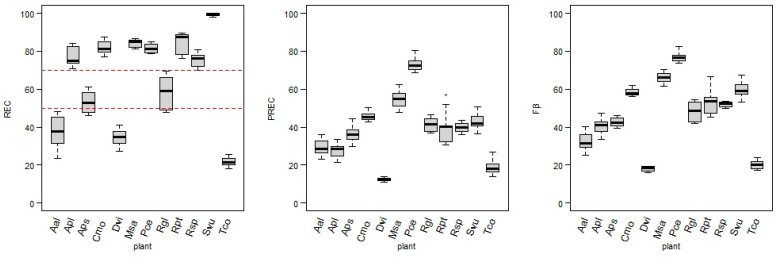
Values of recall (REC) (red dashed lines dividing models into three groups), precision (PREC), and Fβ score (Fβ) obtained for the models of each plant.

**Figure 7 sensors-25-02106-f007:**
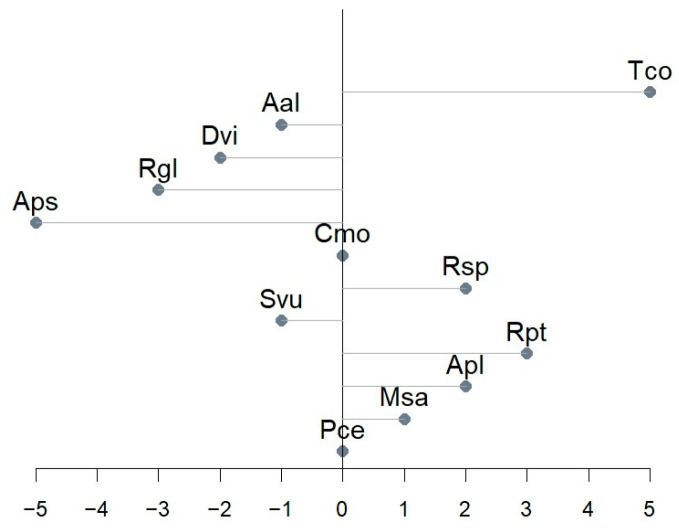
Onset day shift between predicted and designated BBCH 61.

**Table 1 sensors-25-02106-t001:** Plant and flowering characteristics for 12 analyzed woody plants in the camera frame.

Plant Characteristics						Flowering Characteristics			
Code	Species	Family	Form	Distance from Camera [m]	Crown Visibility	Petal Color	Single Flower/Inflorescence Size [cm]	Leaf Presence at Stage BBCH 6	Flower Location on the Shoot
Aal	*Ailanthus altissima*	*Simaroubaceae*	tree	40	full	white green	0.6/30	yes	on top
Apl	*Acer platanoides*	*Sapindaceae*	tree	100	partially covered	yellowgreen	1.2/10	no	on top
Aps	*Acer pseudoplatanus*	*Sapindaceae*	tree	100	partially covered	yellowgreen	0.6/15	yes	under leaves
Cmo	*Crataegus monogyna*	*Rosaceae*	tree	45	partially covered	white	1.5/10	yes	on top
Dvi	*Diospyros virginiana*	*Ebenaceae*	tree	35	full	peach	0.7	yes	under leaves
Msa	*Magnollia salicifolia*	*Magnoliaceae*	tree	25	full	white	10	no	on top
Pce	*Prunus cerasifera*	*Rosaceae*	tree	50	partially covered	white	2	no	on top
Rgl	*Rosa glauca*	*Rosaceae*	shrub	35	full	pink	3	yes	on top
Rpt	*Rosa pteragonis*	*Rosaceae*	shrub	35	full	light yellow	5	yes	on top
Rsp	*Rosa spinosissima*	*Rosaceae*	shrub	35	full	white	4	yes	on top
Svu	*Syringa vulgaris*	*Oleaceae*	shrub	35	full	lilac	0.7/20	yes	on top
Tco	*Tilia cordata*	*Malvaceae*	tree	40	full	cream + yellowgreen bract	1+ bract	yes	under leaves

**Table 2 sensors-25-02106-t002:** Benchmarking Fβ score values for five classification learners (kknn, svm, ranger, naïve bayes, rpart).

	Classification Algorithms				
Plant Code	*kk*nn	svm	Ranger	naive_bayes	rpart
Aal	0.534	0.173	0.493	0.264	-
Apl	0.920	0.887	0.900	0.456	0.791
Aps	0.821	0.711	0.788	0.367	0.675
Cmo	0.932	0.908	0.916	0.251	0.827
Dvi	0.541	-	0.500	0.263	-
Msa	0.952	0.948	0.951	0.683	0.863
Pce	0.889	0.839	0.883	0.573	0.777
Rgl	0.594	0.429	0.571	0.288	0.316
Rpt	0.930	0.906	0.907	0.457	0.839
Rsp	0.892	0.842	0.853	0.277	0.700
Svu	0.973	0.956	0.964	0.645	0.916
Tco	0.456	-	0.373	0.191	-

**Table 3 sensors-25-02106-t003:** Pros and cons of various data treatments applied to class determination.

Treatment	Pros	Cons
A	full dataset from transitional stages	possible observer’s bias
B	class certainty	absence of transitional stages
C	full dataset available; possible label correction for days lacking field observations	possible bias arising from the camera perspective; relying on model performance

**Table 4 sensors-25-02106-t004:** Feature selection results for all 12 plants (✓ stands for features that were most explanatory for the flowering classification and chosen for algorithm training).

	Aal	Apl	Aps	Cmo	Dvi	Msa	Pce	Rgl	Rpt	Rsp	Svu	Tco
*r.av*	✓		✓	✓		✓		✓		✓		
*g.av*		✓				✓			✓		✓	
*b.av*	✓	✓	✓	✓			✓					
*r.sd*	✓	✓	✓	✓		✓			✓	✓		✓
*g.sd*	✓	✓	✓	✓	✓	✓	✓	✓		✓	✓	✓
*b.sd*	✓	✓		✓		✓	✓	✓	✓	✓	✓	✓
*bri.av*			✓		✓	✓	✓	✓		✓		✓
*bri.sd*				✓	✓	✓				✓	✓	
*gi.av*	✓	✓	✓					✓	✓		✓	✓
*gi.sd*	✓	✓	✓		✓	✓	✓	✓		✓	✓	
*gei.av*				✓	✓		✓	✓	✓	✓		
*gei.sd*	✓	✓	✓	✓	✓	✓	✓	✓	✓	✓	✓	✓
*ri.av*		✓			✓	✓	✓	✓				
*ri.sd*	✓		✓	✓	✓			✓				
*bi.av*			✓	✓			✓	✓	✓	✓		✓
*bi.sd*	✓	✓		✓	✓	✓			✓	✓		✓

**Table 5 sensors-25-02106-t005:** Dates of flowering stages BBCH61 and BBCH67 and the flowering duration for each plant in both years of observations.

	2022					2023				
	61		67		Duration [days]	61		67		Duration [days]
	Date	Doy	Date	Doy		Date	Doy	Date	Doy	
Pce	6.04	96	16.04	106	11	3.04	93	17.04	107	15
Msa	7.04	97	28.04	118	22	9.04	99	23.04	113	15
Apl	19.04	109	4.05	124	16	20.04	110	5.05	125	16
Rpt	4.05	124	18.05	138	15	4.05	124	22.05	142	19
Svu	8.05	128	23.05	143	16	9.05	129	25.05	145	17
Rsp	9.05	129	20.05	140	12	12.05	132	24.05	144	13
Cmo	10.05	130	20.05	140	11	14.05	134	25.05	145	12
Aps	16.05	136	24.05	144	9	17.05	137	26.05	146	10
Rgl	27.05	147	13.06	164	18	29.05	149	12.06	163	15
Dvi	11.06	162	21.06	172	11	11.06	162	21.06	172	11
Aal	19.06	170	30.06	181	12	20.06	171	3.07	184	14
Tco	22.06	173	2.07	183	11	24.06	175	4.07	185	11

**Table 6 sensors-25-02106-t006:** Diel classification results for each plant selected model.

Plant	Model	Flowering (BBCH 61–67)			Flowering Extremes			Non-Flowering Stages		
		Observed	Predicted		Observed	Predicted		Observed	Predicted	
		Days	Days	FDS	Days	Days	FDS	Days	Days	FDS
Aal	A.df	12	5	41.7	10	1	10	334	0	0
Apl	B.fs.df	16	14	87.5	12	7	58.3	306	2	0.7
Aps	C.df	9	8	88.9	14	5	35.7	333	3	0.9
Cmo	B.nt/C.nt	12	11	91.7	13	0	0	309	0	0
Dvi	A.df	11	4	36.4	9	1	11.1	336	18	5.4
Msa	B.fs.df	15	14	93.3	16	0	0	303	3	1
Pce	B.df/B.fs.df	15	14	93.3	28	3	10.7	291	0	0
Rgl	C.df	15	13	86.7	12	3	25	307	3	1
Rpt	A.fs.df	19	16	84.2	20	1	5	295	0	0
Rsp	B.fs.df	13	11	84.6	17	0	0	304	0	0
Svu	C.fs	17	17	100	14	5	35.7	303	0	0
Tco	A.df	11	3	27.3	11	0	0	339	0	0

## Data Availability

The raw data supporting the conclusions of this article will be made available by the authors on request.

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
