# Peer review of "Digital Repeat Photography Application for Flowering Stage Classification of Selected Woody Plants"

_sensors, 2025, doi:10.3390/s25072106_

Round 1
Reviewer 1 Report
Comments and Suggestions for Authors
This paper utilizes digital repeat photography and ground-based observation data, which are relatively reliable sources of data with minimal uncertainty. However, there are some issues that need to be addressed:
1.The abstract section lacks specific data, which reduces the reliability of the conclusions. It is recommended to include concrete data in the abstract section to strengthen the conclusions.
2. The results section is overly comprehensive and lacks conciseness. It is suggested to focus on the main research findings and refine the presentation. Additionally, the results section lacks figures. Adding a few figures would make the results more visually intuitive.
3.The discussion section is too lengthy, and some parts deviate from the main research theme. It is recommended to focus the discussion on the main research findings and include more comparisons with other studies. For example, the Potential Applications section should be shortened to one paragraph.
4.The caption for Figure 2 should be placed below the figure.
Author Response
Dear Reviewer,
Thank you for your valuable comments and suggestions. We found them very useful in improving our manuscript. I want to address your every comment in detail:
Comment 1. The abstract section lacks specific data, which reduces the reliability of the conclusions. It is recommended to include concrete data in the abstract section to strengthen the conclusions.
Response 1. Thank you for your valuable comments and suggestions. We found them very useful in improving our manuscript. We have changed the abstract: taking your advice, we added specific data related to the comparison of our method and conventional phenological observations [lines 26-29]. We slightly changed the sentence order and added one sentence about methods [22-26] for more appropriate abstract composition.
Comment 2. The results section is overly comprehensive and lacks conciseness. It is suggested to focus on the main research findings and refine the presentation. Additionally, the results section lacks figures. Adding a few figures would make the results more visually intuitive.
Response 2. As You suggested the description of field results was shortened and described differently [393-419]. On section 3.2. Classification efficiency [420-448] we added one figure (Figure 6) and one table in Appendix B. Section 3.3 Diel classifications were rearranged [454-497]
Comment 3.The discussion section is too lengthy, and some parts deviate from the main research theme. It is recommended to focus the discussion on the main research findings and include more comparisons with other studies. For example, the Potential Applications section should be shortened to one paragraph.
Response 3. Discussion section is been rearranged almost completely, as there were more comments on its length and structure [498-725]. Potential Applications section is changed [702-725]. Section 4.3 Phenological methods comparison [638-701] brings comparisons with other studies, with one new added.
Comment 4.The caption for Figure 2 should be placed below the figure.
Response 4. The caption of figure 2 (now Figure 3, page 9) is corrected.
Reviewer 2 Report
Comments and Suggestions for Authors
This paper introduces a fresh approach that combines digital repeat photography with a weighted k-nearest neighbors algorithm to automatically identify when various woody plants are flowering. The authors use a long-term series of high-frequency images along with field observations to train their models. They explore three different data treatments and several transformation techniques to improve the model’s performance. Overall, the study aims to address challenges related to plant morphology and environmental effects on phenological observations.
The authors provide a very detailed description of their methodology, which includes clear steps for preprocessing, applying three distinct data treatments, and various data transformations to boost model performance.
The detailed breakdown of the three data treatments (observer’s perspective, uncertain data removal, and semi-supervised learning) is informative but can feel a bit overwhelming. It might help readers if this section were streamlined or summarized in a table that clearly outlines the pros and cons of each approach.
I appreciate that the authors compared several algorithms (like SVM, random forest, etc.) before choosing weighted knn, and that they supported their choice with benchmark results. The emphasis on recall is understandable given the unbalanced dataset; however, I would like to see a more detailed discussion on why recall was prioritized over precision or F1. More insight into misclassification issues, especially for the species with lower performance, would also add value.
Combining traditional field observations with remote imagery is a strong point of this work, and it’s great to see these methods used together to validate the automated predictions. The paper could benefit from a more in-depth discussion on the subjectivity of ground observations. For instance, it might be useful to discuss how observer bias could influence the training data and what could be done to reduce this bias in future studies.
The manuscript does a good job explaining the experimental workflow from data collection to model tuning. For others to reproduce this work, providing additional details on the hyperparameter tuning (like the selection of k and kernel functions) and the exact experimental setup would be very helpful. Including a supplementary section or a link to a repository with the code could greatly enhance reproducibility.
The discussion provides a thoughtful look at the limitations of the current approach, such as the effects of plant morphology and environmental variability on classification accuracy. This section could be more concise. Separating the interpretation of results, the limitations, and the ideas for future research into clear sub-sections might help the reader follow the narrative more easily. Also, discussing the broader implications for automated phenological monitoring could strengthen the paper’s impact.
Some paragraphs are quite long and might be easier to digest if broken down into bullet points or shorter sections.
Make sure every figure and table is clearly referenced in the text. Simplifying tables that report performance metrics could improve clarity.
A careful proofreading to address minor grammatical issues and streamline the language would benefit the overall readability.
It would be useful to briefly compare this approach with other remote sensing methods (like satellite-based monitoring) in terms of resolution, cost, and practical application.
Author Response
Dear Reviewer,
Thank you for your valuable comments and suggestions. Your kind words gave us strength to work on revision. We found them very useful in improving our manuscript. I want to address your every comment in detail:
Comment 1.
The detailed breakdown of the three data treatments (observer’s perspective, uncertain data removal, and semi-supervised learning) is informative but can feel a bit overwhelming. It might help readers if this section were streamlined or summarized in a table that clearly outlines the pros and cons of each approach.
Response 1.
Data treatments A, B, C as now described graphically as well (Figure 2., page 8) and little table about pros and cons of every treatment is attached (Table 3., page 8).
Comment 2.
I appreciate that the authors compared several algorithms (like SVM, random forest, etc.) before choosing weighted knn, and that they supported their choice with benchmark results. The emphasis on recall is understandable given the unbalanced dataset; however, I would like to see a more detailed discussion on why recall was prioritized over precision or F1. More insight into misclassification issues, especially for the species with lower performance, would also add value.
Response 2.
In section 2.4.2. Algorithm selection [lines 231- 236], we provided some additional arguments for choosing recall as a dominant measure in this study.
Comment 3.
Combining traditional field observations with remote imagery is a strong point of this work, and it’s great to see these methods used together to validate the automated predictions. The paper could benefit from a more in-depth discussion on the subjectivity of ground observations. For instance, it might be useful to discuss how observer bias could influence the training data and what could be done to reduce this bias in future studies.
Response 3.
In discussion, section 4.2 Limitations, 4.2.2. Observer’s bias [lines 595-627], we added some information about ground observation subjectivity. In the last sentences of this section we added some helpful tips improving the observations [630-632].
Comment 4.
The manuscript does a good job explaining the experimental workflow from data collection to model tuning. For others to reproduce this work, providing additional details on the hyperparameter tuning (like the selection of k and kernel functions) and the exact experimental setup would be very helpful. Including a supplementary section or a link to a repository with the code could greatly enhance reproducibility.
Response 4.
In section 2.4.6. [334-351] more details are added about hyperparameters tuning. Due to time consuming revisions we did not have time to figure out the best way to put make our codes avaiable but we'll try to in upcomming days.
Comment 5.
The discussion provides a thoughtful look at the limitations of the current approach, such as the effects of plant morphology and environmental variability on classification accuracy. This section could be more concise. Separating the interpretation of results, the limitations, and the ideas for future research into clear sub-sections might help the reader follow the narrative more easily. Also, discussing the broader implications for automated phenological monitoring could strengthen the paper’s impact.
Response 5. Thank you very much for this comment. We rearranged the Disscusion section (starting from page 15) due to your suggestion and it seems more sorted out. Now it is divided into a description of models performance, method limitations, phenological method comparison (section changed a little due to other reviewer comment), and potential applications.
Comment 6.
Some paragraphs are quite long and might be easier to digest if broken down into bullet points or shorter sections.
Response 6.
We tried to keep paragraphs short while making our revisions, and we hope the current layout is more readable.
Comment 7.
A careful proofreading to address minor grammatical issues and streamline the language would benefit the overall readability.
Response 7.
The whole text was checked again for some language mistakes, we're hope this time manuscript have less errors of this kind.
Comment 8.
It would be useful to briefly compare this approach with other remote sensing methods (like satellite-based monitoring) in terms of resolution, cost, and practical application.
Response 8.
In lines 679-688 we added short paragraph about method comparison.
Reviewer 3 Report
Comments and Suggestions for Authors
The manuscript applies the digital repeat photography technique to the phenological observation and classification of woody plants, showcasing good innovation in the research. The study indicates that Phenocameras can enhance the efficiency of conventional phenological research. The paper is well-written and logically clear. Recommendation for acceptance. Some modifications needed in the paper are as follows:
1. In the "Materials and Methods" section of Part Two, methods for evaluating classification accuracy need to be supplemented. The relevant content is missing in the paper, so please provide the formulas for the relevant indicators.
2 Please supplement the advantages and disadvantages of "The digital repeat photography technique" in the discussion section
Author Response
Dear Reviewer,
Thank you for your valuable comments and suggestions. We found them very useful in improving our manuscript. I want to address your every comment in detail:
Comment 1. In the "Materials and Methods" section of Part Two, methods for evaluating classification accuracy need to be supplemented. The relevant content is missing in the paper, so please provide the formulas for the relevant indicators.
Response 1. In section test datasets [lines 357-389] we added suggested formulas. Indeed the text was not complete without them.
Comment 2. Please supplement the advantages and disadvantages of "The digital repeat photography technique" in the discussion section
Response 2. In section 4.3. Phenological methods comparison [lines 679-688] we added some brief informations about pros and cons of digital repeat photography.
Round 2
Reviewer 1 Report
Comments and Suggestions for Authors
The remarks and concerns are well addressed in the revision. I recommend this manuscript for publication.